# Don't take it lightly: Phasing optical random projections with unknown operators

**Sidharth Gupta**
University of Illinois at Urbana-Champaign
gupta67@illinois.edu

**Rémi Gribonval**
Univ Rennes, Inria, CNRS, IRISA
remi.gribonval@inria.fr

**Laurent Daudet**
LightOn, Paris
laurent@lighton.ai

**Ivan Dokmanić**
University of Illinois at Urbana-Champaign
dokmanic@illinois.edu

## Abstract

In this paper we tackle the problem of recovering the phase of complex linear measurements when only magnitude information is available and we control the input. We are motivated by the recent development of dedicated optics-based hardware for rapid random projections which leverages the propagation of light in random media. A signal of interest $\boldsymbol{\xi} \in \mathbb{R}^N$ is mixed by a random scattering medium to compute the projection $\boldsymbol{y} = \boldsymbol{A}\boldsymbol{\xi}$, with $\boldsymbol{A} \in \mathbb{C}^{M \times N}$ being a realization of a standard complex Gaussian iid random matrix. Such optics-based matrix multiplications can be much faster and energy-efficient than their CPU or GPU counterparts, yet two difficulties must be resolved: only the intensity $|\boldsymbol{y}|^2$ can be recorded by the camera, and the transmission matrix $\boldsymbol{A}$ is unknown. We show that even without knowing $\boldsymbol{A}$, we can recover the unknown phase of $\boldsymbol{y}$ for some *equivalent* transmission matrix with the same distribution as $\boldsymbol{A}$. Our method is based on two observations: first, conjugating or changing the phase of any row of $\boldsymbol{A}$ does not change its distribution; and second, since we control the input we can interfere $\boldsymbol{\xi}$ with arbitrary reference signals. We show how to leverage these observations to cast the *measurement phase retrieval problem* as a Euclidean distance geometry problem. We demonstrate appealing properties of the proposed algorithm in both numerical simulations and real hardware experiments. Not only does our algorithm accurately recover the missing phase, but it mitigates the effects of quantization and the sensitivity threshold, thus improving the measured magnitudes.

## 1 Introduction

Random projections are at the heart of many algorithms in machine learning, signal processing and numerical linear algebra. Recent developments ranging from classification with random features [16], kernel approximation [25] and sketching for matrix optimization [24, 27], to sublinear-complexity transforms [26] and randomized linear algebra are all enabled by random projections. Computing random projections for realistic signals such as images, videos, and modern big data streams is computation- and memory-intensive. Thus, from a practical point of view, any increase in the size and speed at which one can do the required processing is highly desirable.

This fact has motivated work on using dedicated hardware based on physics rather than traditional CPU and GPU computation to obtain random projections. A notable example is the scattering of light in random media (Figure 1 (left)) with an optical processing unit (OPU). The OPU enables rapid (20 kHz) projections of high-dimensional data such as images, with input dimension scaling up

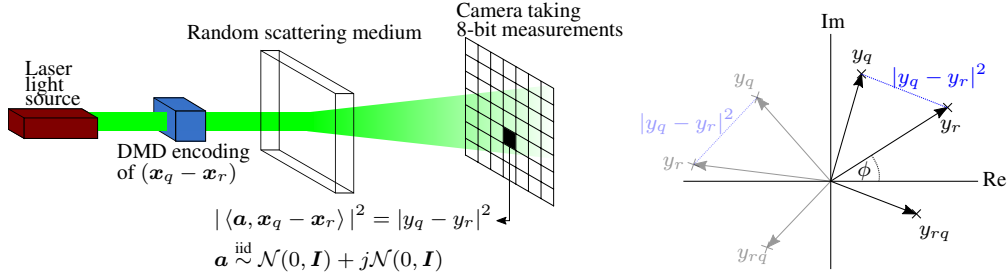

Figure 1: Left: The optical processing unit (OPU) is an example application of where the MPR problem appears. A coherent laser beam spatially encodes a signal $(\boldsymbol{x}_q - \boldsymbol{x}_r)$ via a digital micro-mirror device (DMD) which is then shined through a random medium. A camera measures the squared magnitude of the scattered light which is equivalent to the Euclidean distance between complex numbers $y_q \in \mathbb{C}$ and $y_r \in \mathbb{C}$. Furthermore the camera takes quantized measurements; Right: $y_q$ and $y_r$ are points on the two-dimensional complex plane. We can measure the squared Euclidean distance between points and use these distances to localize points on the complex plane and obtain their phase. Note that transformations such as rotations and reflections do not change the distances.

to one million and output dimension also in the million range. It works by "imprinting" the input data $\boldsymbol{\xi} \in \mathbb{R}^N$ onto a coherent light beam using a digital micro-mirror device (DMD) and shining the modulated light through a multiple scattering medium such as titanium dioxide white paint. The scattered lightfield in the sensor plane can then be written as

$$\boldsymbol{y} = \boldsymbol{A}\boldsymbol{\xi}$$

where $\boldsymbol{A} \in \mathbb{C}^{M \times N}$ is the transmission matrix of the random medium with desirable properties.

One of the major challenges associated with this approach is that $\boldsymbol{A}$ is in general *unknown*. Though it could in principle be learned via calibration [6], such a procedure is slow and inconvenient, especially at high resolution. On the other hand, the system can be designed so that the distribution of $\boldsymbol{A}$ is approximately iid standard complex Gaussian. Luckily, this fact alone is sufficient for many algorithms and the actual values of $\boldsymbol{A}$ are not required.

Another challenge is that common light sensors are only sensitive to intensity, so we can only measure the intensity of scattered light, $|\boldsymbol{y}|^2$, where $|\cdot|$ is the elementwise absolute value. The phase information is thus lost. While the use of interferometric measurements with a reference could enable estimating the phase, the practical setup is more complex, sensitive, and it does not share the convenience and simplicity of the one illustrated in Figure 1 (left).

This motivates us to consider the *measurement phase retrieval (MPR) problem*. The MPR sensor data is modeled as

$$\boldsymbol{b} = |\boldsymbol{y}|^2 + \boldsymbol{\eta} = |\boldsymbol{A}\boldsymbol{\xi}|^2 + \boldsymbol{\eta}, \tag{1}$$

where $\boldsymbol{b} \in \mathbb{R}^M$, $\boldsymbol{\xi} \in \mathbb{R}^N$, $\boldsymbol{A} \in \mathbb{C}^{M \times N}$, $\boldsymbol{y} \in \mathbb{C}^M$, and $\boldsymbol{\eta} \in \mathbb{R}^M$ is noise. The goal is to recover the phase of each complex-valued element of $\boldsymbol{y}$, $y_i$ for $1 \leq i \leq M$, from its magnitude measurements $\boldsymbol{b}$ when $\boldsymbol{\xi}$ is *known* and the entries of $\boldsymbol{A}$ are *unknown*. The classical phase retrieval problem which has received much attention over the last decade [15, 4] has the same quadratic form as (1) but with a known $\boldsymbol{A}$ and the task being to recover $\boldsymbol{\xi}$ instead of $\boldsymbol{y}$. While at a glance it might seem that not knowing $\boldsymbol{A}$ precludes computing the phase of $\boldsymbol{A}\boldsymbol{\xi}$, we show in this paper that it is in fact possible via an exercise in distance geometry.

The noise $\boldsymbol{\eta}$ is primarily due to quantization because standard camera sensors measure low precision values, 8-bit in our case (integers between 0 and 255 inclusive). Furthermore, cameras may perform poorly at low intensities. This is another data-dependent noise source which is modelled in (2) by a binary mask vector $\boldsymbol{w} \in \mathbb{R}^M$ which is zero when the intensity is below some threshold and one otherwise; $\odot$ denotes the elementwise product.

$$\boldsymbol{b} = \boldsymbol{w} \odot \left( |\boldsymbol{y}|^2 + \boldsymbol{\eta} \right) = \boldsymbol{w} \odot \left( |\boldsymbol{A}\boldsymbol{\xi}|^2 + \boldsymbol{\eta} \right) \tag{2}$$

The distribution of $\boldsymbol{A}$ follows from the properties of random scattering media [14, 6]. It has iid standard complex Gaussian entries, $a_{mn} \sim \mathcal{N}(0, 1) + j\mathcal{N}(0, 1)$ for all $1 \leq m, n \leq M, N$.

The usefulness of phase is obvious. While in some applications having only the magnitude of the random projection is enough (see [17] for an example related to elliptic kernels), most applications require the phase. For example, with the phase one can implement a more diverse range of kernels as well as randomized linear algebra routines like randomized singular value decomposition (SVD). We report the results of the latter on real hardware in Section 3.1.

**Our contributions.** We develop an algorithm based on distance geometry to solve the MPR problem (1). We exploit the fact that we control the input to the system, which allows us to mix $\boldsymbol{\xi}$ with arbitrary reference inputs. By interpreting each pixel value as a point in the complex plane, this leads to a formulation of the MPR problem as a pure distance geometry problem (see Section 2.2 and Figure 1 (right)). With enough pairwise distances (corresponding to reference signals) we can localize the points on the complex plane via a variant of multidimensional scaling (MDS) [23, 5], and thus compute the missing phase.

As we demonstrate, the proposed algorithm not only accurately recovers the phase, but also improves the number of useful bits of the magnitude information thanks to the multiple views. Established Euclidean distance geometry bounds imply that even with many distances below the sensitivity threshold and coarse quantization, the proposed algorithm allows for accurate recovery. This fact, which we verify experimentally, could have bearing on the design of future random projectors by navigating the tradeoff between physics and computation.

## 1.1 Related work

The classical phase retrieval problem looks at the case where $\boldsymbol{A}$ is known and $\boldsymbol{\xi}$ has to be recovered from $\boldsymbol{b}$ in (1) [7, 21, 10]. A modified version of the classical problem known as holographic phase retrieval is related to our approach: a known reference signal is concatenated with $\boldsymbol{\xi}$ to facilitate the phase estimation [1]. Interference with known references for classical phase retrieval has also been studied for known (Fourier) operators [3, 11] .

An optical random projection setup similar to the one we consider has been used for kernel-based classification [17], albeit using only magnitudes. A phaseless approach to classification with the measured magnitudes fed into a convolutional neural network was reported by Satat et al. [18].

An alternative to obtaining the measurement phase is to measure, or calibrate, the unknown transmission matrix $\boldsymbol{A}$. This has been attempted in compressive imaging applications but the process is impractical at even moderate pixel counts [6, 14]. Estimating $\boldsymbol{A}$ can take days and even the latest GPU-accelerated methods take hours for moderately sized $\boldsymbol{A}$ [20]. Other approaches forego calibration and use the measured magnitudes to *learn* an inverse map of $\boldsymbol{x} \mapsto |\boldsymbol{A}\boldsymbol{x}|^2$ for use with the magnitude measurements [9].

Leaving hardware approaches aside, there have been multiple algorithmic efforts to improve the speed of random projections [12, 25] for machine learning and signal processing tasks. Still, efficiently handling high-dimensional input remains a formidable challenge.

## 2 The measurement phase retrieval problem

We will denote the signal of interest by $\boldsymbol{\xi} \in \mathbb{R}^N$, and the $K$ reference *anchor* signals by $\boldsymbol{r}_k \in \mathbb{R}^N$ for $1 \leq k \leq K$. To present the full algorithm we will need to use multiple signals of interest which we will then denote $\boldsymbol{\xi}_1, \ldots, \boldsymbol{\xi}_S$; each $\boldsymbol{\xi}_s$ is called a frame. We set the last, $K$th anchor to be the origin, $\boldsymbol{r}_K = \boldsymbol{0}$. We ascribe $\boldsymbol{\xi}$ and the anchors to the columns of the matrix $\boldsymbol{X} \in \mathbb{R}^{N \times Q}$, so that $\boldsymbol{X} = [\boldsymbol{\xi}, \boldsymbol{r}_1, \boldsymbol{r}_2, \cdots, \boldsymbol{r}_K]$ and let $Q = K + 1$. The $q$th column of $\boldsymbol{X}$ is denoted $\boldsymbol{x}_q$. For any $1 \leq q, r \leq Q$, we let $\boldsymbol{y}_q = \boldsymbol{A}\boldsymbol{x}_q$ and $\boldsymbol{y}_{qr} := \boldsymbol{A}(\boldsymbol{x}_q - \boldsymbol{x}_r)$, with $y_{qr,m}$ being its $m$th entry. Finally, the $m$th row of $\boldsymbol{A}$ will be denoted by $\boldsymbol{a}^m$ so that $y_{qr,m} = \langle \boldsymbol{a}^m, \boldsymbol{x}_q - \boldsymbol{x}_r \rangle$.

## 2.1 Problem statement and recovery up to a reference phase and conjugation

Since we do not know $\boldsymbol{A}$, it is clear that recovering the absolute phase of $\boldsymbol{A}\boldsymbol{\xi}$ is impossible. On the other hand, many algorithms do not require any knowledge of $\boldsymbol{A}$ except that it is iid standard complex Gaussian, and that it does not change throughout the computations.

Let R be an operator which adds a constant phase to each row of its argument (multiplies it by $\text{diag}(e^{j\phi_1}, \ldots, e^{j\phi_m})$ for some $\phi_1, \ldots, \phi_m$) and conjugates a subset of its rows. Since a standard complex Gaussian is circularly symmetric, $\mathsf{R}(\boldsymbol{A})$ has the same distribution as $\boldsymbol{A}$. Therefore, since we do not know $\boldsymbol{A}$, it does not matter whether we work with $\boldsymbol{A}$ itself or with $\mathsf{R}(\boldsymbol{A})$ for some possibly unknown R. As long as the same *effective* R is used for all inputs during algorithm operation, the relative phases between the frames will be the same whether we use $\mathsf{R}(\boldsymbol{A})$ or $\boldsymbol{A}$.[1]

**Problem 1.** *Given a collection of input frames $\boldsymbol{\xi}_1, \ldots, \boldsymbol{\xi}_S$ to be randomly projected and a device illustrated in Figure 1 (left) with an unknown transmission matrix $\boldsymbol{A} \in \mathbb{C}^{M \times N}$ and a b-bit camera, compute the estimates of projections $\hat{\boldsymbol{y}}_1, \ldots, \hat{\boldsymbol{y}}_S$ up to a global row-wise phase and conjugation; that is, so that there exists some R such that $\hat{\boldsymbol{y}}_s \approx \mathsf{R}(\boldsymbol{y}_s)$ for all $1 \le s \le S$.*

## 2.2 MPR as a distance geometry problem

Since the rows of $\boldsymbol{A}$ are statistically independent, we can explain our algorithm for a single row and then repeat the same steps for the remaining rows. We will therefore omit the row subscript/superscript $m$ except where explicitly necessary.

Instead of randomly projecting $\boldsymbol{\xi}$ and measuring the corresponding projection magnitude $|\boldsymbol{A}\boldsymbol{\xi}|^2$, consider randomly projecting the difference between $\boldsymbol{\xi}$ and some reference vector, or more generally a difference between two columns in $\boldsymbol{X}$, thus measuring $|\langle \boldsymbol{a}, \boldsymbol{x}_q - \boldsymbol{x}_r \rangle|^2 = |y_q - y_r|^2$. Interpreting $y_q$ and $y_r$ as points in the complex plane, we see that the camera sensor measures exactly the squared Euclidean distance between them. Since we control the input to the OPU, we can indeed set it to $\boldsymbol{x}_q - \boldsymbol{x}_r$ and measure $|y_q - y_r|^2$ for all $1 \le q, r \le Q$.

This is the key point: as we can measure pairwise distances between a collection of two-dimensional vectors in the two-dimensional complex plane, we can use established distance geometry algorithms such as multidimensional scaling (MDS) to localize points and get their phase. This is illustrated in Figure 1 (right). The same figure also illustrates the well known fact that rigid transformations of a point set cannot be recovered from distance data. We need to worry about three things: translations, reflections and rotations.

The translation ambiguity can be easily dealt with if one notes that for any column $\boldsymbol{x}_q$ of $\boldsymbol{X}$, $|y_q| = |\langle \boldsymbol{a}, \boldsymbol{x}_q \rangle|$ gives us the distance of $y_q$ to the origin which is a fixed point, ultimately resolving the translation ambiguity. There is, however, no similar simple way to do away with the rotation and reflection ambiguity, so it might seem that there is no way to uniquely determine the phase of $\langle \boldsymbol{a}, \boldsymbol{\xi} \rangle$. This is where the discussion from the preceding subsection comes to the rescue. Since R is arbitrary, as long as it is kept fixed for all the frames, we can arbitrarily set the orientation of any given frame and use it as a reference, making sure that the relative phases are computed correctly.

## 2.3 Proposed algorithm

As defined previously, the columns of $\boldsymbol{X} \in \mathbb{R}^{N \times Q}$ list the signal of interest and the anchors. Recall that all the entries of $\boldsymbol{X}$ are known. Using the OPU, we can compute a noisy (quantized) version of

$$|y_{qr}|^2 = |\langle \boldsymbol{a}, \boldsymbol{x}_q - \boldsymbol{x}_r \rangle|^2 = |y_q - y_r|^2, \tag{3}$$

for all $(q, r)$, which gives us $Q(Q-1)/2$ squared Euclidean distances between points $\{y_q \in \mathbb{C}\}_{q=1}^{Q}$ on the complex plane. These distances can be used to populate a Euclidean (squared) distance matrix $\boldsymbol{D} \in \mathbb{R}^{Q \times Q}$ as $\boldsymbol{D} = (d_{qr}^2)_{q,r=1}^{Q} = (|y_{qr}|^2)_{q,r=1}^{Q}$, which we will use to localize all complex points $y_q$.

We start by defining the matrix of all the complex points in $\mathbb{R}^2$ which we want to recover as

$$\boldsymbol{\Upsilon} = \begin{bmatrix} \text{Re}(y_1) & \text{Re}(y_2) & \cdots & \text{Re}(y_Q) \\ \text{Im}(y_1) & \text{Im}(y_2) & \cdots & \text{Im}(y_Q) \end{bmatrix} \in \mathbb{R}^{2 \times Q}.$$

Denoting the $q$th column of $\boldsymbol{\Upsilon}$ by $\boldsymbol{v}_q$, we have $d_{qr}^2 = \|\boldsymbol{v}_q - \boldsymbol{v}_r\|_2^2 = \boldsymbol{v}_q^T \boldsymbol{v}_q - 2\boldsymbol{v}_q^T \boldsymbol{v}_r + \boldsymbol{v}_r^T \boldsymbol{v}_r$ so that

$$\boldsymbol{D} = \text{diag}(\boldsymbol{G}) \mathbf{1}_Q^T - 2\boldsymbol{G} + \mathbf{1}_Q \text{diag}(\boldsymbol{G})^T =: \mathcal{K}(\boldsymbol{G}), \tag{4}$$

where $\operatorname{diag}(\boldsymbol{G}) \in \mathbb{R}^Q$ is the column vector of the diagonal entries in the Gram matrix $\boldsymbol{G} := \boldsymbol{\Upsilon}^T \boldsymbol{\Upsilon} \in \mathbb{R}^{Q \times Q}$ and $\mathbf{1}_Q \in \mathbb{R}^Q$ is the column vector of $Q$ ones. This establishes a relationship between the measured distances in $\boldsymbol{D}$ and the locations of the complex points in $\mathbb{R}^2$ which we seek. We denote by $\boldsymbol{J}$ the geometric centering matrix, $\boldsymbol{J} := \boldsymbol{I} - \frac{1}{Q}\mathbf{1}_Q\mathbf{1}_Q^T$ so that

$$\widehat{\boldsymbol{G}} = -\tfrac{1}{2}\boldsymbol{J}\boldsymbol{D}\boldsymbol{J} = \boldsymbol{J}\boldsymbol{G}\boldsymbol{J} = (\boldsymbol{\Upsilon}\boldsymbol{J})^T(\boldsymbol{\Upsilon}\boldsymbol{J}) \tag{5}$$

is the Gram matrix of the centered point set in terms of $\boldsymbol{\Upsilon}$. $\widehat{\boldsymbol{G}}$ and $\boldsymbol{J}$ are know as the Gram matrix of the centered point set and the geometric centering matrix because $\boldsymbol{\Upsilon}\boldsymbol{J}$ is the points in $\boldsymbol{\Upsilon}$ with their mean subtracted. An estimate $\widehat{\boldsymbol{\Upsilon}}$ of the centered point set, $\boldsymbol{\Upsilon}\boldsymbol{J}$, is then obtained by eigendecomposition as $\widehat{\boldsymbol{G}} = \boldsymbol{V}\operatorname{diag}(\lambda_1, \ldots, \lambda_Q)\boldsymbol{V}^T$ and taking $\widehat{\boldsymbol{\Upsilon}} = [\sqrt{\lambda_1}\boldsymbol{v}_1, \sqrt{\lambda_2}\boldsymbol{v}_2]^T$ where $\boldsymbol{v}_1$ and $\boldsymbol{v}_2$ are the first and second columns of $\boldsymbol{V}$ and assuming that the eigenvalue sequence is nonincreasing. This process is the classical MDS algorithm [23, 5]. Finally, the phases can be calculated via a four-quadrant inverse tangent, $\phi(y_q) = \arctan(v_{q2}, v_{q1})$.

**Procrustes analysis.**    As we recovered a centered point set via MDS with a geometric centering matrix $\boldsymbol{J}$, the point set will have its centroid at the origin. This is a consequence of the used algorithm, and not the "true" origin. As described above, we know that $|y_q|^2$ defines squared distances to the origin and $y_Q = \langle \boldsymbol{a}, \boldsymbol{x}_Q \rangle = 0 + 0j$ (as $\boldsymbol{x}_Q$ was set to the origin), meaning that we can correctly center the recovered points by translating the point set, $\widehat{\boldsymbol{\Upsilon}}$, by $-\boldsymbol{v}_Q$.

The correct absolute rotation and reflection cannot be recovered. However, since we only care about working with some effective $\mathsf{R}(\boldsymbol{A})$ with the correct distribution, we only need to ensure that the relative phases between the frames are correct. We can thus designate the first frame as the reference frame and set the rotation (which directly corresponds to the phase) and reflection (corresponding to conjugation) arbitrarily. Once these are chosen, the anchors $\boldsymbol{r}_1, \ldots, \boldsymbol{r}_K$ are fixed, which in turn fixes the phasing–conjugation operator $\mathsf{R}$.

Since $\boldsymbol{A}$ is unknown, $\mathsf{R}$ is also unknown, but fixed anchors allow us to compute the correct relative phase with respect to $\mathsf{R}(\boldsymbol{A})$ for the subsequent frames. Namely, upon receiving a new input $\boldsymbol{\xi}_s$ to be randomly projected, we now localize it with respect to a fixed set of anchors. This is achieved by Procrustes analysis. Denoting by $\widetilde{\boldsymbol{\Upsilon}}_1$ our reference estimate of the anchor positions in frame 1 (columns $2, \ldots, Q$ of $\widehat{\boldsymbol{\Upsilon}}$ above which was recovered from $\widehat{\boldsymbol{G}}$ in (5)), and by $\widetilde{\boldsymbol{\Upsilon}}_s$ the MDS estimate of anchor positions in frame $s$, adequately centered. Let $\widetilde{\boldsymbol{\Upsilon}}_s\widetilde{\boldsymbol{\Upsilon}}_1^T = \boldsymbol{U}\boldsymbol{\Sigma}\boldsymbol{V}^T$ be the singular value decomposition of $\widetilde{\boldsymbol{\Upsilon}}_s\widetilde{\boldsymbol{\Upsilon}}_1^T$. The optimal transformation matrix in the least squares sense is then $\boldsymbol{R} = \boldsymbol{V}\boldsymbol{U}^T$ so that $\boldsymbol{R}\widetilde{\boldsymbol{\Upsilon}}_s \approx \widetilde{\boldsymbol{\Upsilon}}_1$ [19].

Finally, we note that with a good estimate of the anchors, one can imagine not relocalizing them in every frame. The localization problem for $\boldsymbol{\xi}$ then boils down to multilateration, cf. Section C in the supplementary material.

## 2.4   Sensitivity threshold and missing measurements

As we further elaborate in Section A of the supplementary material, in practice some measurements fall below the sensitivity threshold of the camera and produce spurious values. A nice benefit of multiple "views" of $\boldsymbol{\xi}$ via its interaction with reference signals is that we can ignore those measurements. This introduces missing values in $\boldsymbol{D}$ which can be modeled via a binary mask matrix $\boldsymbol{W}$. The recovery problem can be modeled as estimating $\boldsymbol{\Upsilon}$ from $\boldsymbol{W} \odot (\boldsymbol{D} + \boldsymbol{E})$ where $\boldsymbol{W} \in \mathbb{R}^{N \times N}$ contains zeros for the entries which fall below some prescribed threshold, and ones otherwise.

We can predict the performance of the proposed method when modeling the entries of $\boldsymbol{W}$ as iid Bernoulli random variables with parameter $p$, where $1 - p$ is the probability that an entry falls below the sensitivity threshold and $\boldsymbol{E}$ as uniform quantization noise distributed as $\mathcal{U}\left(-\frac{\kappa}{2(2^b-1)}, \frac{\kappa}{2(2^b-1)}\right)$, where $b$ is the number of bits, and $\kappa$ an upper bound on the entries of $\boldsymbol{D}$ (in our case $2^8 - 1 = 255$).

Adapting existing results on the performance of multidimensional scaling [28] (by noting that $\boldsymbol{E}$ is sub-Gaussian), we can get the following scaling of the distance recovery error with the number of

---
**Algorithm 1** MPR algorithm for $S$ frames.

---

**Input:** Squared distances $\left[\left|y_{jQ,m} - y_{lQ,m}\right|^2\right]_s$ for all $1 \leq j, l \leq Q$ for frames $1 \leq s \leq S$ and rows $1 \leq m \leq M$; $[\cdot]_s$ denotes frame $s$

**Output:** $\boldsymbol{Y} \in \mathbb{C}^{M \times S}$ containing all localized points such that $\boldsymbol{y}_s = \mathsf{R}(\boldsymbol{A})\boldsymbol{\xi}_s$ for some fixed $\mathsf{R}$.

1:  $\boldsymbol{Y} \leftarrow \boldsymbol{0}_{M \times S}$                                                   $\triangleright$ Initialize $\boldsymbol{Y}$
2:  $m \leftarrow 1$
3: **while** $m \leq M$ **do**                                 $\triangleright$ Solve each row separately
4:     Populate all frame $s = 1$ distances into distance matrix $\boldsymbol{D}$          $\triangleright$ $\boldsymbol{D} \in \mathbb{R}^{Q \times Q}$
5:     $[\boldsymbol{\Upsilon}]_1 \leftarrow \mathtt{MDS}(\boldsymbol{D})$                                   $\triangleright$ $[\boldsymbol{\Upsilon}]_1 \in \mathbb{R}^{2 \times Q}$
6:     $[\boldsymbol{\Upsilon}]_1 \leftarrow \mathtt{GradientDescent}(\boldsymbol{D}, [\boldsymbol{\Upsilon}]_1)$
7:     $[\boldsymbol{\Upsilon}]_1 \leftarrow [\boldsymbol{\Upsilon}]_1 - [\boldsymbol{v}_Q]_1 \boldsymbol{1}^T$                     $\triangleright$ Translate to align with origin
8:     $s \leftarrow 2$
9:     **while** $s \leq S$ **do**
10:        Populate all frame $s$ distances into distance matrix $\boldsymbol{D}$
11:        $[\boldsymbol{\Upsilon}]_s \leftarrow \mathtt{MDS}(\boldsymbol{D})$
12:        $[\boldsymbol{\Upsilon}]_s \leftarrow \mathtt{GradientDescent}(\boldsymbol{D}, [\boldsymbol{\Upsilon}]_s)$
13:        $[\boldsymbol{\Upsilon}]_s \leftarrow [\boldsymbol{\Upsilon}]_s - [\boldsymbol{v}_Q]_s \boldsymbol{1}^T$
14:        $\boldsymbol{R} \leftarrow \mathtt{Procrustes}([\boldsymbol{v}_2, \dots, \boldsymbol{v}_Q]_1, [\boldsymbol{v}_2, \dots, \boldsymbol{v}_Q]_s)$   $\triangleright$ $\boldsymbol{R}$ aligns frames 1 and $s$ anchors
15:        $[\boldsymbol{\Upsilon}]_s \leftarrow \mathtt{Align}([\boldsymbol{\Upsilon}]_s, \boldsymbol{R}, [\boldsymbol{v}_2, \dots, \boldsymbol{v}_Q]_1)$              $\triangleright$ Align anchors
16:        $s \leftarrow s + 1$
17:     **end while**
18:     $\boldsymbol{U} \leftarrow \left[[\boldsymbol{v}_1]_1, [\boldsymbol{v}_1]_2, \dots, [\boldsymbol{v}_1]_S\right]$                         $\triangleright$ $\boldsymbol{U} \in \mathbb{R}^{2 \times S}$
19:     $\boldsymbol{y}^m \leftarrow \boldsymbol{u}^1 + j\boldsymbol{u}^2$           $\triangleright$ Multiply second row of $\boldsymbol{U}$ with $j$ and add to first row
20:     $m \leftarrow m + 1$
21: **end while**

---

anchors $K$ (for $K$ sufficiently large),

$$\frac{1}{K}\mathbb{E}\left[\left\|\hat{\boldsymbol{D}} - \boldsymbol{D}\right\|_F\right] \lesssim \frac{\kappa}{\sqrt{pK}}, \tag{6}$$

where $\lesssim$ denotes inequality up to a constant which depends on the number of bits $b$, the sub-Gaussian norm of the entries in $\boldsymbol{E}$, and the dimension of the ambient space (here $\mathbb{R}^2$). An important implication is that even for coarse quantization (small $b$) and for a large fraction of entries below the sensitivity threshold (small $p$), we can achieve arbitrarily small amplitude and phase errors *per point* by increasing the number of reference signals $K$.

**Refinement with gradient descent.** The output of the classical MDS method described above can be further refined via a local search. A standard differentiable objective called the squared stress is defined as follows,

$$\min_{\boldsymbol{Z}} f(\boldsymbol{\Upsilon}) = \min_{\boldsymbol{Z}} \left\|\boldsymbol{W} \odot \left(\boldsymbol{D} - \mathcal{K}\left(\boldsymbol{Z}^T\boldsymbol{Z}\right)\right)\right\|_F^2, \tag{7}$$

where $\mathcal{K}(\cdot)$ is as defined in (4) and $\boldsymbol{Z} \in \mathbb{R}^{2 \times Q}$ is the point matrix induced by row $m$ of $\boldsymbol{A}$. In our experiments we report the result of refining the classical MDS results via gradient descent on (7).

Note that the optimization (7) is nonconvex. The complete procedure is thus analogous to the usual approach to nonconvex phase retrieval by spectral initialization followed by gradient descent [15, 4]. Algorithm 1 summarizes our proposed method.

## 3   Experimental verification and application

We test the proposed MPR algorithm via simulations and experiments on a real OPU. For hardware experiments, we use a scikit-learn interface to a publicly available cloud-based OPU.[2]

Reproducible code available at `https://github.com/swing-research/opu_phase` under the MIT License.

**Evaluation metrics.** The main challenge is to evaluate the performance without knowing the transmission matrix $\boldsymbol{A}$. To this end, we propose to use the *linearity error*. The rationale behind this metric is that with the phase correctly recovered, the end-to-end system should be linear. That is, if we recover $\boldsymbol{y}$ and $\boldsymbol{z}$ from $|\boldsymbol{y}|^2 = |\boldsymbol{A}\boldsymbol{\xi}_1|^2$ and $|\boldsymbol{z}|^2 = |\boldsymbol{A}\boldsymbol{\xi}_2|^2$, then we should get $(\boldsymbol{y} + \boldsymbol{z})$ when applying the method to $|\boldsymbol{v}|^2 = |\boldsymbol{A}(\boldsymbol{\xi}_1 + \boldsymbol{\xi}_2)|^2$. With this notation, the relative linearity error is defined as

$$\text{linearity error} = \frac{1}{M} \sum_{m=1}^{M} \frac{|(y_m + z_m) - v_m|}{|v_m|}. \tag{8}$$

The second metric we use is the number of "good" or correct bits. This metric can only be evaluated in simulation since it requires the knowledge of the ground truth measurements. Letting $|y|^2 = |\langle \boldsymbol{a}, \boldsymbol{\xi} \rangle|^2$ and $\hat{y}$ be our estimate of $y$, the number of good bits is defined as

$$\text{good bits} = -\frac{20}{6.02} \log \left( \left| |y|^2 - |\hat{y}|^2 \right| / |y|^2 \right).$$

It is proportional to the signal-to-quantization-noise ratio if the distances uniformly cover all quantization levels.[3]

## 3.1 Experiments

In all simulations, intensity measurements are quantized to 8 bits and all signals and references are iid standard (complex) Gaussian random vectors.

We first test the phase recovery performance by evaluating the linearity error. In simulation, we draw random frames $\boldsymbol{\xi}_1$, $\boldsymbol{\xi}_2$, and $\boldsymbol{A} \in \mathbb{C}^{100 \times 64^2}$. We apply Algorithm 1 to $|\boldsymbol{A}\boldsymbol{\xi}_1|^2$, $|\boldsymbol{A}\boldsymbol{\xi}_2|^2$ and $|\boldsymbol{A}(\boldsymbol{\xi}_1 + \boldsymbol{\xi}_2)|^2$ and calculate the linearity error (8). We use classical MDS and MDS with gradient descent (MDS-GD). Figure 2a shows that the system is indeed approximately linear and that the linearity error becomes smaller as the number of reference signals grows. In Figure 2b, we set the sensitivity threshold to $\tau = 6$ and zero the distances below the threshold per (2). Again, the linearity error quickly becomes small as the number of anchors increases showing that the overall system is robust and that it allows recovery of phase for small-intensity signals.

Next, we test the linearity error with a real hardware OPU. The OPU gives 8-bit unsigned integer measurements. A major challenge is that the DMD (see Figure 1) only allows binary input signals. This is a property of the particular OPU we use and while it imposes restrictions on reference design, the method is unchanged as our algorithm does not assume a particular type of signal. Section A in the supplementary material describes how we create binary references and addresses other hardware-related practicalities.

Figure 2c reports the linearity error on the OPU with suitably designed references and the same size $\boldsymbol{A}$. The empirically determined sensitivity threshold of the camera is $\tau = 6$, and the measurements below the threshold were not used. We ignore rows of $\boldsymbol{A}$ which give points with small norms (less than two) because they are prone to noise and disproportionately influence the relative error. Once again, we observe that the end-to-end system with Algorithm 1 is approximately linear and that the linearity improves as we increase the number of anchors.

Finally, we demonstrate the magnitude denoising performance. We draw $\boldsymbol{a} \in \mathbb{C}^{100}$, a random signal $\boldsymbol{\xi} \in \mathbb{R}^{100}$ and a set of random reference anchor signals. We run our algorithm for number of anchors varying between 2 and 15. For each number of anchors, we recover $\hat{y}$ for $|y|^2 = |\langle \boldsymbol{a}, \boldsymbol{\xi} \rangle|^2$ using either classical MDS or MDS-GD. We then measure the number of good bits. The average results over 100 trials are shown in Figure 3a. Figure 3b reports the same experiment with the sensitivity threshold set to $\tau = 6$ (that is, the entries below $\tau$ are zeroed in the distance matrix per (2)). Both figures show that the proposed algorithm significantly improves the estimated magnitudes in addition to recovering the phases. The approximately 1 additional good bit with gradient descent in Figure 3b corresponds to the relative value of $2^1/2^8 \approx 0.8\%$ which is consistent with the gradient descent improvement in Figure 2b.

We also test a scenario where the anchor positions on the complex plane are known exactly and we only have to localize a single measurement. We compare this to localizing the anchors and the

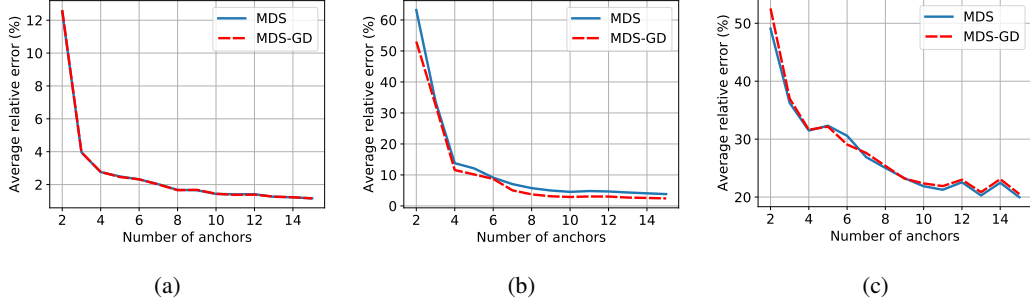

(a)            (b)            (c)

Figure 2: Experiments in simulation and on real hardware to evaluate the linearity error as defined in (8). The input signals are of dimension $64^2$, $M$ in (8) is 100 and the number of anchors signals are increased. The classical MDS and MDS with gradient descent (MDS-GD) are used. In all cases the error decreases as the number of anchors increases. (a) In simulation with Gaussian signals and Gaussian reference signals; (b) In simulation with Gaussian signals and Gaussian reference signals with sensitivity threshold $\tau = 6$; (c) On a real OPU with binary signals and binary references.

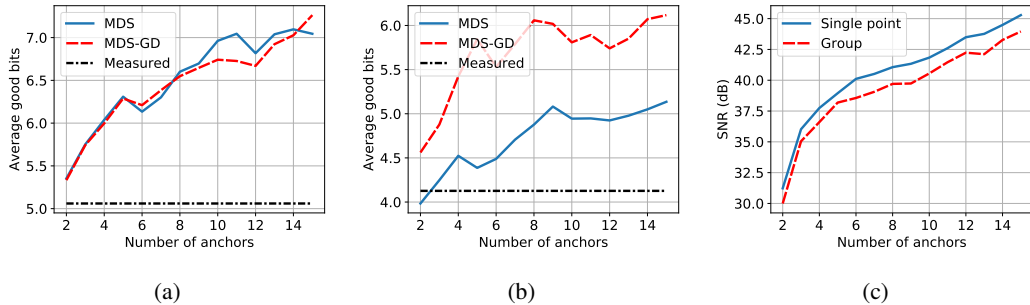

(a)            (b)            (c)

Figure 3: (a) Magnitude denoising performance of MDS and MDS-GD over 100 trials. Input signals are Gaussian and of dimension 100; (b) Magnitude denoising performance of MDS and MDS-GD over 100 trials with 100-dimensional Gaussian signals and sensitivity threshold $\tau = 6$; (c) Comparison between recovering a single point and recovering the point and anchors at the same time. SR-LS is used to locate a single point when anchors are known and MDS is used to locate all points when anchors are unknown.

measurements jointly. Localizing a single point via multilateration is performed by minimizing the SR-LS objective (see (9) in the supplementary material). The input signal dimension is $64^2$ and we recover $\hat{y}$ for $|y|^2 = |\langle \boldsymbol{a}, \boldsymbol{\xi} \rangle|^2$. We perform 100 trials and calculate the SNR of the recovered complex points. Figure 3c shows that although having perfect knowledge of anchor locations helps, classical MDS alone does not perform much worse.

**Optical randomized singular value decomposition.** We use Algorithm 1 to implement randomized singular value decomposition (RSVD) as described in Halko et al. [8] on the OPU. We use 5 anchors in all RSVD experiments. The original RSVD algorithm and a variant with adaptations for the OPU are described in Algorithms 2 and 3 in the supplementary material.

One of the steps in the RSVD algorithm for an input matrix $\boldsymbol{B} \in \mathbb{R}^{M \times N}$ requires the computation of $\boldsymbol{B\Omega}$ where $\boldsymbol{\Omega} \in \mathbb{R}^{N \times 2K}$ is a standard real Gaussian matrix, $K$ is the target number of singular vectors, and $2K$ may be interpreted as the number of random projections for each row of $\boldsymbol{B}$. We use the OPU to compute this random matrix multiplication. An interesting observation is that since in Algorithm 1 we recover the result of multiplications by a complex matrix with independent real and imaginary parts, we can halve the number of projections when using the OPU with respect to the original algorithm. By treating each row of $\boldsymbol{B}$ as an input frame, we can obtain $\boldsymbol{Y} \in \mathbb{C}^{K \times M}$ via Algorithm 1 when $|\boldsymbol{Y}|^2 = |\boldsymbol{A}\boldsymbol{B}^T|^2$ with $\boldsymbol{A}$ as defined in Problem 1 with $K$ rows. Then, we can construct $\boldsymbol{P} = [\mathrm{Re}(\boldsymbol{Y}^*) \quad \mathrm{Im}(\boldsymbol{Y}^*)] \in \mathbb{R}^{M \times 2K}$ which would be equivalent to computing $\boldsymbol{B\Omega}$ for real $\boldsymbol{\Omega}$. Section B in the supplementary material describes this in more detail.

Figure 4 shows the results when the OPU is used to perform the random matrix multiplication of the RSVD algorithm on a matrix $\boldsymbol{B}$. Figure 4 (left) reports experiments with a random binary matrix $\boldsymbol{B} \in \mathbb{R}^{10 \times 10^4}$, different numbers of random projections (number of rows in $\boldsymbol{A}$), and ten trials per number of projections. We plot the average error per entry when reconstructing $\boldsymbol{B}$ from its RSVD matrices and singular values. Next, we take 500 $28 \times 28$ samples from the MNIST dataset [13], threshold them to be binary, vectorize them, and stack them into a matrix $\boldsymbol{B} \in \mathbb{R}^{500 \times 28^2}$. Figure 4 (right) shows the seven leading right singular vectors reshaped to $28 \times 28$. The top row shows the singular vectors that are obtained when using the OPU with 500 projections and the bottom row shows the result when using Python. The error is negligible.

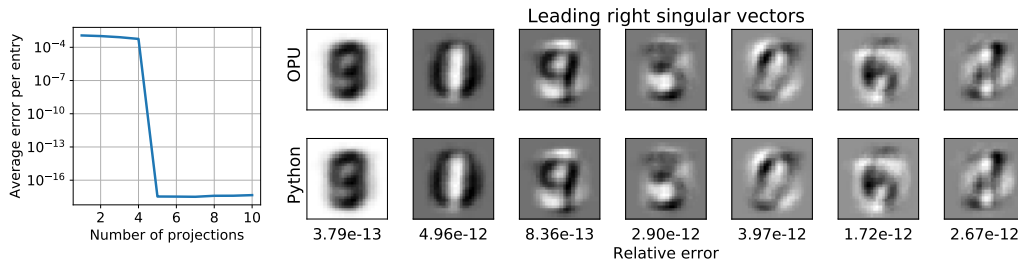

Figure 4: Left: Average RSVD error over 10 trials with varying number of projections on hardware with an input matrix of size $10 \times 1000$; Right: Reshaped leading right singular vectors of an MNIST matrix of size $500 \times 28^2$. The top rows shows the leading right singular vectors after performing RSVD with the OPU and using our algorithm. The bottom row shows the leading right singular vectors from Python. The relative error is below each singular vector.

## 4 Conclusion

Traditional computation methods are often too slow for processing tasks which involve large data streams. This motivates alternatives which instead use fast physics to "compute" the desired functions. In this work, we looked at using optics and multiple scattering media to obtain linear random projections. A common difficulty with optical systems is that off-the-shelf camera sensors only register the intensity of the scattered light. Our results show that there is nevertheless no need to reach for more complicated and more expensive coherent setups. We showed that measurement phase retrieval can be cast as a problem in distance geometry, and that the unknown phase of random projections can be recovered even without knowing the transmission matrix of the medium.

Simulations and experiments on real hardware show that the OPU setup combined with our algorithm indeed approximates an end-to-end linear system. What is more, we also improve intensity measurements. The fact that we get full complex measurements allows us to implement a whole new spectrum of randomized algorithms; we demonstrated the potential by the randomized singular value decomposition. These benefits come at the expense of a reduction in data throughput. Future work will have to precisely quantify the smallest achievable data rate reduction due to allocating a part of the duty cycle for reference measurements, though we note that the optical processing data rates are very high to begin with.

## Acknowledgement

Sidharth Gupta and Ivan Dokmanić would like to acknowledge support from the National Science Foundation under Grant CIF-1817577.

## Footnotes

[1]Up to a sign.

[2]`https://www.lighton.ai/lighton-cloud/`.

[3]Note that the quantity registered by the camera is actually the squared magnitude, hence the factor 20.

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
