[Supplementary Material]

# Supplementary material

## A  Practical considerations with hardware

**Reference design.**  On the OPU system described in Figure 1, the DMD only lets us encode and randomly project binary signals. Therefore, all pairwise differences $(\boldsymbol{x}_q - \boldsymbol{x}_r)$ between the columns of $\boldsymbol{X} \in \mathbb{R}^{N \times Q}$ must be binary. To design reference signals we first collect all frames $\{\boldsymbol{\xi}_s\}_{s=1}^S$ and sum them $\sum_{s=1}^S \boldsymbol{\xi}_s$. The first reference $\boldsymbol{r}_1$ is initialized to ones at the indices where $\sum_{s=1}^S \boldsymbol{\xi}_s$ is nonzero. Next, some of $\boldsymbol{r}_1$'s zero-valued entries are flipped to one with probability $\alpha$. A similar process is used for all subsequent references. In general, a reference $\boldsymbol{r}_q$ is initialized by assigning ones to the nonzero support of $\left( \sum_{s=1}^S \boldsymbol{\xi}_s + \sum_{k=1}^{q-1} \boldsymbol{r}_k \right)$ and then flipping some of its zero entries with probability $\alpha$.

There is a tradeoff between large and small $\alpha$. If $\alpha$ is too large, a reference may become all-ones before all subsequent references are generated. On the other hand if $\alpha$ is too small, $\boldsymbol{r}_{q+1} - \boldsymbol{r}_q$ will have many zeros and so $|\boldsymbol{A}(\boldsymbol{r}_{q+1} - \boldsymbol{r}_q)|^2$ may not be high enough to be detected by the camera sensor. The consequence of this tradeoff is that in practice the number of anchors is limited. Furthermore, in general a larger $N$ makes it easier to make $K$ good anchors as $\alpha$ can be larger which keeps $|\boldsymbol{A}(\boldsymbol{r}_{q+1} - \boldsymbol{r}_q)|^2$ away from the sensitivity threshold.

Figure 5 shows binary references reshaped into squares which were used for the linearity experiment on the OPU in Figure 2c. Here, $N = 64^2$ and $\alpha = 0.2$. The number on top of each reference is the difference in the number of ones between itself and the previously generated reference.

Figure 5: Binary references reshaped into squares which were used for the linearity experiment on the OPU in Figure 2c. Here $N = 64^2$ and $\alpha = 0.2$. The number on top of each anchor is the difference in the number of ones between itself and the previously generated reference.

**Minimum attainable measurement.**  To determine the sensitivity threshold, $\tau$, we randomly project an all-zero signal a few times and record the output. The minimum, mode or mean of these measurements can be taken to be the minimum that can be measured. Once $\tau$ is estimated, we apply a mask and zero any measurements which are equal to or less than the minimum.

**Camera sensor saturation.**  It is possible for the signal reaching the camera to saturate the sensor. In a $b$-bit system we can detect this if many measurements are equal to $2^b - 1$. In all experiments we ensure that there is no saturation by adjusting the camera exposure. This again involves a tradeoff: if exposure is too high, we saturate the sensor; if it is too low, measurements may be too small to be detected and we are not exploiting the full dynamic range.

# B Randomized singular value decomposition (RSVD) details

Algorithm 2 is the prototype randomized SVD algorithm given by [8]. To implement this on hardware we replace Step 1 and 2 to formulate Algorithm 3. As $\boldsymbol{A}$ in Algorithm 3 has iid entries following a standard complex Gaussian, calculating $\boldsymbol{P}$ in Algorithm 3 is the same as doing step 2 in Algorithm 2. We only need to do half the number of projections because we use an iid complex random matrix. The real and imaginary parts are two random projections.

---

**Algorithm 2** Prototype randomized SVD algorithm [8].

---

**Input:** Matrix, $\boldsymbol{B} \in \mathbb{R}^{M \times N}$ whose SVD is required, a target number of $K$ singular vectors
**Output:** The SVD $U$, $\boldsymbol{\Sigma}$ and $\boldsymbol{V}^*$
 1: Generate an $N \times 2K$ random Gaussian matrix $\boldsymbol{A}$.
 2: Form $\boldsymbol{Y} = \boldsymbol{B}\boldsymbol{A}$.
 3: Construct a matrix $\boldsymbol{Q}$ whose columns form an orthonormal basis for the range of $\boldsymbol{Y}$.
 4: Form $\boldsymbol{C} = \boldsymbol{Q}^* \boldsymbol{B}$.
 5: Compute the SVD of the smaller $\boldsymbol{C} = \tilde{\boldsymbol{U}} \boldsymbol{\Sigma} \boldsymbol{V}^*$.
 6: $\boldsymbol{U} = \boldsymbol{Q}\tilde{\boldsymbol{U}}$.

---

**Algorithm 3** Randomized SVD algorithm on the OPU.

---

**Input:** Matrix, $\boldsymbol{B} \in \mathbb{R}^{M \times N}$ whose SVD is required, a target number of $K$ singular vectors
**Output:** The SVD $U$, $\boldsymbol{\Sigma}$ and $\boldsymbol{V}^*$
 1: Solve $|\boldsymbol{Y}|^2 = |\boldsymbol{A}\boldsymbol{B}^*|^2$ by treating each column of $\boldsymbol{B}^*$ as a frame and using Algorithm 1, where $\boldsymbol{A} \in \mathbb{C}^{K \times N}$ is as in the MPR problem and has $K$ rows.
 2: Horizontally stack the real and imaginary parts of $\boldsymbol{Y}^* \in \mathbb{C}^{M \times K}$ as $\boldsymbol{P} = [\text{Re}(\boldsymbol{Y}^*) \quad \text{Im}(\boldsymbol{Y}^*)] \in \mathbb{R}^{M \times 2K}$.
 3: Construct a matrix $\boldsymbol{Q}$ whose columns form an orthonormal basis for the range of $\boldsymbol{P}$.
 4: Form $\boldsymbol{C} = \boldsymbol{Q}^* \boldsymbol{B}$.
 5: Compute the SVD of the smaller $\boldsymbol{C} = \tilde{\boldsymbol{U}} \boldsymbol{\Sigma} \boldsymbol{V}^*$.
 6: $\boldsymbol{U} = \boldsymbol{Q}\tilde{\boldsymbol{U}}$.

---

# C Localization with known anchor positions

If we have perfect knowledge of the anchor locations in the complex plane, we do not need to localize them for each frame $s$. The localization problem then boils down to multilateration, which can be formulated by minimizing the square-range-based least squares (SR-LS) objective [2],

$$\widehat{\boldsymbol{v}}_1 = \min_{\boldsymbol{v}_1} \sum_{q=2}^{Q} \left( \|\boldsymbol{v}_1 - \boldsymbol{v}_q\|_2^2 - d_q^2 \right)^2 \tag{9}$$

where $\boldsymbol{v}_1$ and $\boldsymbol{v}_q$ are as defined in Section 2.3 and $d_q$ is the noisy measured distance. There exists efficient algorithms which solve (9) to global optimality [2], as well as suboptimal solutions based on solving a small linear system [22].