[Reviews · NeurIPS 2019]

Reviewer 1



The paper is generally well-written and provides useful illustrations. It is novel and I believe technically correct. It provides an important result for a small field. Comments: It's worth noting that computationally efficient methods to measure TMs have been developed [A]. [A] Sharma, Manoj, et al. "Inverse Scattering via Transmission Matrices: Broadband Illumination and Fast Phase Retrieval Algorithms." IEEE Transactions on Computational Imaging (2019). I had not seen the use of a definition on the rhs of an equation before, as is done in (4). This made the text a little difficult to parse. Typos: Line 80: "Experimantly" Line 277: "ore"

Reviewer 2



Originality: The problem formulation is interesting. The main part of the algorithm relies on the well-known multidimensional scaling (MDS) and procrustes methods. Quality: The quality of work presented in this paper seems good and the authors tried to explain the problem clearly. Adding some more details on the experiments would be helpful. Clarity: The paper is well written. Significance: Phase retrieval problem itself is considered an interesting and challenging research problem that arises in many applications. The case where the random operator is also unknown makes it a more challenging problem, but I am not aware of many scenarios where such a problem arises. The authors motivated the problem with OPU setup, but the results presented in the paper are on a very small scale and not very convincing about potential advantages of the proposed framework. In short, I like the problem on the algorithmic level, but I am doubtful about the practical significance.

Reviewer 3



The paper is written in a confusing way that the setup is not explained clearly. There is no clear reason to me why there need to be multiple signals -- since the measurements are taken on all difference of pairs x_q-x_r. The setting of multiple signals also seems to have nothing to do with the described MDS algorithm. [The author responded that one cannot determine the global phase with one signal -- that is true, but the authors have auxiliary vectors r_1,...,r_K and why do they need s \xi vectors?] The organization is not good, either. Assumptions such as “DMD is binary” seems to be popped up all of a sudden from nowhere. The paper also cited some other algorithm without explaining what the problems other algorithms solve. It would be more readable if the authors can reorganize the paper and explain more clearly in the motivation part/problem setup what the constraints are due to hardware limitations. The authors further mention that they use the algorithms developed in this paper to implement the RSVD algorithm in [8]. I think they can be more specific here. What the authors actually implement is to compute A = BG for some given matrix B and a Gaussian random matrix A. [The authors seems to suggest in their response that it is a harder problem to do the matrix-vector multiplication simulation simultaneously for multiple signals, but why is it so? Is it due to hardware constraints?] It seems that the point is to use an optical system to simulate the computation of A = BG via some decoding algorithm. What is the advantage of this? [The authors responded that an optimal system is much faster than both CPU and GPU. Perhaps that should be made more explicit in the paper. Neither the introduction nor the experiment mentions speed comparison.]

[Author Response · NeurIPS 2019]

We thank the reviewers for taking the time to read and comment on our paper. Below we first group the comments of Reviewers #1 and #2 by topic, and then address the more critical concerns of Reviewer #3.

**Scope of applications.** Reviewers #1 and #2 find the work novel and interesting, but comment that it might cater to a small community. While our focus on optical random projections could suggest so, the methodology is widely applicable: our method linearizes any imaging, computation, or communication system in which the phase is important but only the intensity is measured, regardless of whether the transmission matrix (TM) is known or unknown. While mostly optical, these include X-ray and terahertz imaging, electron lasers, etc. The key is to have some control over the input. We give two concrete, relevant examples related to reference [A] suggested by Reviewer #1 (thank you!):

1) Once the TM of the medium is calibrated, imaging still requires phase retrieval. We present the method for an unknown $A$, but it applies just the same with $A$ known. If one introduces a set of interferers in the scene whose response is known, then the phase can be obtained rapidly via our proposed method;
2) More interestingly, our method reduces TM learning to a *linear* system (vs quadratic). The same Algorithm 1 determines the phase of the measurements, yielding a linear system for every row of $A$. Finally, unknown TMs remain relevant, as calibration is slow and many media do change over time. We stop here for brevity, but indeed feel that the method can have a broad impact. The above examples can be easily incorporated into the manuscript.

**Suggestions for improvement.** The reviewers suggested to improve the clarity of Section 2.3 and experiment and figure descriptions, which we are happy to do. Reviewer #2 also commented as follows:

*1) The related work does not clearly show the gap.* We feel that the cited work puts our work in context with the existing literature: Tasks which do not recover the measurement phase [9, 17, 18] have limited performance. Transmission matrix estimation [6, 14] is computationally heavy for large resolution / dynamic media. Our algorithm recovers the phase in real time since it only requires computation of small SVDs, e.g., $6 \times 6$.
*2) Difference in performance between MDS and MDS-GD.* Fig. 2b and 3b use different metrics (relative error vs good bits); note that a 1 bit difference which is clearly visible in 3b amounts to only 0.8% relative error. The difference between Fig. 3a and 3b is that in 3a there is only quantization, while in 3b there is thresholding, hence larger bit error.

**Reviewer #3.** The reviewer summarizes our work as developing an algorithm for the measurement phase retrieval (MPR) problem in an OPU. Rather, we propose a general algorithm for the MPR, OPU being an example application.

**Multiple signals and binary inputs.** The reviewer writes that the paper is confusing, questioning the need for multiple frames. We explain this in l. 134-140: in short, for a single input $\xi_1$, since the matrix $A$ is unknown (and so the global phase of each row), we can choose the phase of $\langle a_i, \xi_1 \rangle$ arbitrarily and the MPR is vacuous (cf. rotation invariance of complex Gaussians, l. 108-113). We thus use the first frame as a reference and align future frames with it via Procrustes analysis (l. 157-172), yielding correct *relative* phases. The reviewer next states that "*The supp. mat. says that the problem in the main text considers only binary signals.*" We respectfully disagree. The main text considers general signals, and simulation experiments use general signals as evident from l. 216-226, 237-249. Only the particular hardware we use for *real-world experiments* has a binary DMD. This is standard and due to the use of cheap, fast modulators developed for movie projectors, as opposed to the slow and costly liquid crystal tech.

The reviewer then writes "*The organization is not good, either. Assumptions such as 'DMD is binary' seems to be popped up all of a sudden from nowhere.*" Again, we find this surprising since it appears only on p. 7 when we show results with real hardware (instead of stopping at numerics): "*Next, we test the linearity error with a real hardware OPU. The OPU gives 8-bit unsigned integer measurements. A major challenge is that since the DMD is binary, the inputs need to be binary.*". This does not influence the organization of the paper nor the algorithm. It merely affects reference design. The reviewer continues "*The paper also cited some other algorithm without explaining what the problems other algorithms solve.*" Unfortunately we are not sure what this refers to. (Perhaps: the pseudocode for RSVD is given in the supplement; the algorithm is standard.)

**Randomized SVD.** As RSVD uses multiple random projections, our phase estimates must be consistent over many frames (l. 108-113, Problem 1), hence this is more challenging than a matrix–vector product. This experiment shows that the linearization is good enough for real-life applications. To answer the reviewer's question "*Perhaps the authors just think this is a fun supplementary discovery?*": This is not the case. Showing that our phases are sufficiently good for RSVD *on real hardware* is a strong demonstration of the method's practicality in a context relevant for machine learning and scientific computing. Benchmarks for standard and randomized SVD on standard hardware, and optical randomized SVD would be interesting, but those will be very much hardware-dependent and therefore out of the scope of this paper. With large-enough inputs, optical computation inevitably becomes faster than both CPU and GPU, as its timing is *essentially independent* of the input size. To give an order of magnitude, the optical unit we use is faster than GPUs for input/output dimensions above a few thousands—more than the presented MNIST problem, but much below the size of real-life problems. For example, `Tropp et al., Streaming Low-Rank Matrix Approximation with an Application to Scientific Simulation.` applies a variant of RSVD to climate data of up to 75 GB (!).

[Meta-Review · NeurIPS 2019]

This paper initially received borderline scores, but aided by the authors' response and an active reviewer discussion period, it was decided that the paper should be accepted. I urge the authors to take the reviewers' comments and suggestions into careful consideration when editing their paper to its final version. This paper is on a rather niche topic for NeurIPS, so it will be important to make the work and contributions as clear as possible to non-experts who read the paper.